# CT-Net: Channel Tensorization Network for Video Classification

**Kunchang Li**[12]*, **Xianhang Li**[14]*, **Yali Wang**[13]*, **Jun Wang**[4] **& Yu Qiao**[13]†

[1]Guangdong-Hong Kong-Macao Joint Laboratory of Human-Machine
Intelligence-Synergy Systems, Shenzhen Institutes of Advanced Technology,
Chinese Academy of Sciences, Shenzhen, 518055, China
[2]University of Chinese Academy of Sciences
[3]SIAT Branch, Shenzhen Institute of Artificial Intelligence and Robotics for Society
[4]University of Central Florida
{kc.li, yl.wang, yu.qiao}@siat.ac.cn
xianhangli@knights.ucf.edu, Jun.Wang@ucf.edu

## ABSTRACT

3D convolution is powerful for video classification but often computationally expensive, recent studies mainly focus on decomposing it on spatial-temporal and/or channel dimensions. Unfortunately, most approaches fail to achieve a preferable balance between convolutional efficiency and feature-interaction sufficiency. For this reason, we propose a concise and novel Channel Tensorization Network (CT-Net), by treating the channel dimension of input feature as a multiplication of $K$ sub-dimensions. On one hand, it naturally factorizes convolution in a multiple dimension way, leading to a light computation burden. On the other hand, it can effectively enhance feature interaction from different channels, and progressively enlarge the 3D receptive field of such interaction to boost classification accuracy. Furthermore, we equip our CT-Module with a Tensor Excitation (TE) mechanism. It can learn to exploit spatial, temporal and channel attention in a high-dimensional manner, to improve the cooperative power of all the feature dimensions in our CT-Module. Finally, we flexibly adapt ResNet as our CT-Net. Extensive experiments are conducted on several challenging video benchmarks, e.g., Kinetics-400, Something-Something V1 and V2. Our CT-Net outperforms a number of recent SOTA approaches, in terms of accuracy and/or efficiency.

## 1 INTRODUCTION

3D convolution has been widely used to learn spatial-temporal representation for video classification (Tran et al., 2015; Carreira & Zisserman, 2017). However, over parameterization often makes it computationally expensive and hard to train. To alleviate such difficulty, recent studies mainly focus on decomposing 3D convolution (Tran et al., 2018; 2019). One popular approach is spatial-temporal factorization (Qiu et al., 2017; Tran et al., 2018; Xie et al., 2018), which can reduce overfitting by replacing 3D convolution with 2D spatial convolution and 1D temporal convolution. But it still introduces unnecessary computation burden, since both spatial convolution and temporal convolution are performed over all the feature channels. To further decrease such computation cost, channel separation has been recently developed via operating 3D convolution in the depth-wise manner (Tran et al., 2019). However, it inevitably loses accuracy due to the lack of feature interaction between different channels. For compensation, it has to introduce point-wise convolution to preserve interaction with extra computation. So there is a natural question: *How to construct effective 3D convolution to achieve a preferable trade-off between efficiency and accuracy for video classification?*

---

*Equally-contributed first authors ({kc.li, yl.wang}@siat.ac.cn,
xianhangli@knights.ucf.edu)
†Corresponding author (yu.qiao@siat.ac.cn)

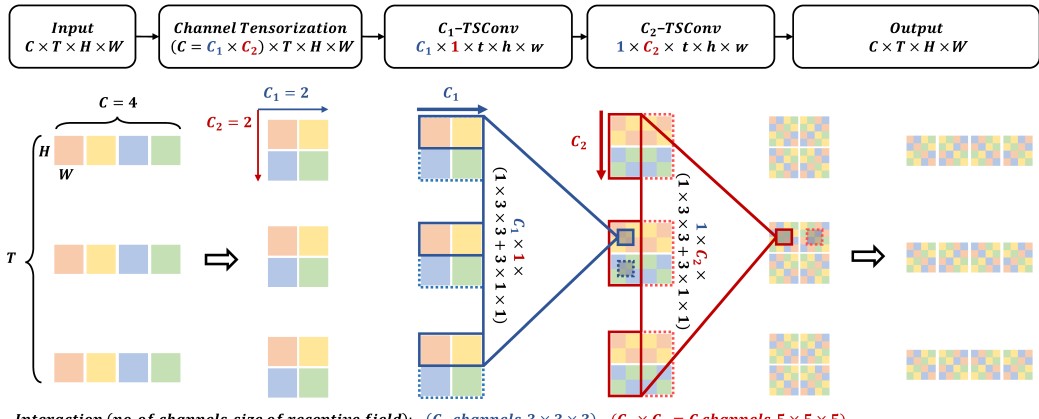

Figure 1: Simple illustration of channel tensorization ($K = 2$). We tensorize the channel dimension of input feature as a multiplication of $K$ sub-dimensions. Via performing spatial/temporal tensor separable convolution along each sub-dimension, we can achieve a preferable balance between convolutional efficiency and feature-interaction sufficiency. Introduction shows more explanations.

| Method | 3D Convolution ($t \times h \times w$) | Convolutional Efficiency | | Feature-Interaction Sufficiency | |
|---|---|---|---|---|---|
| | | Spatial-temporal | Channel | Interact Manner | Interact Field[1] |
| C3D (Tran et al., 2015) | $Full:\ 3 \times 3 \times 3$ | ✗ | ✗ | $STC$ | $3^3$ |
| R(2+1)D (Tran et al., 2018) | $Full:\ 1 \times 3 \times 3$ $Full:\ 3 \times 1 \times 1$ | ✔ | ✗ | $SC$ $TC$ | $3^3$ |
| CSN (Tran et al., 2019) | $Full:\ 1 \times 1 \times 1$ $DW:\ 3 \times 3 \times 3$ | ✗ | ✔ | $C$ $ST$ | $3^3$ |
| Our CT-Net ($C = C_1 \times \cdots \times C_K$) | $C_1:\ C_1 \times \cdots \times 1 \times (1 \times 3 \times 3\ +\ 3 \times 1 \times 1)$ $C_K:\ 1 \times \cdots \times C_K \times (1 \times 3 \times 3\ +\ 3 \times 1 \times 1)$ | ✔ | ✔ | $STC_1$ $STC_K$ | $(2K+1)^3$ |

[1] *Interact Field* means the receptive field for feature interaction.

Table 1: Two design principles to build effective video representation and efficient convolution.

This paper attempts to address this question by investigating two design principles. **(1) Convolutional Efficiency**. As shown in Table 1, current designs of spatial-temporal convolution mainly focus on decomposition from either spatial-temporal (Tran et al., 2018) or channel dimension (Tran et al., 2019). To enhance convolutional efficiency, we consider decomposing convolution in a higher dimension with a novel representation of feature tensor. **(2) Feature-Interaction Sufficiency**. Table 1 clearly shows that, for current decomposition approaches (Tran et al., 2018; 2019), feature interaction only contains one or two of spatial, temporal and channel dimensions at each sub-operation. Such a partial interaction manner would reduce classification accuracy. On one hand, it decreases the discriminative power of video representation, due to the lack of joint learning on all the dimensions. On the other hand, it restricts feature interaction in a limited receptive field, which ignores rich context from a larger 3D region. Hence, to boost classification accuracy, each sub-operation should achieve feature interaction on all the dimensions, and the receptive field of such interaction should be progressively enlarged as the number of sub-operations increases.

Based on these desirable principles, we design a novel and concise Channel Tensorization Module (CT-Module). Specifically, we propose to tensorize the channel dimension of input feature as a multiplication of $K$ sub-dimensions, i.e., $C = C_1 \times C_2 \times \cdots \times C_K$. Via performing spatial/temporal separable convolution along each sub-dimension, we can effectively achieve convolutional efficiency and feature-interaction sufficiency. For better understanding, we use the case of $K = 2$ as a simple illustration in Figure 1. First, we tensorize the input channel into $C = C_1 \times C_2$. Naturally, we separate the convolution into distinct ones along each sub-dimension, e.g., for the $1^{st}$ sub-dimension, we apply our spatial-temporal tensor separable convolution with the size $C_1 \times 1 \times t \times h \times w$, which allows us to achieve convolutional efficiency on all the spatial, temporal and channel dimensions. After that, we sequentially perform the tensor separable convolution sub-dimension by sub-dimension. As a result, we can progressively achieve feature interaction on all the channels, and enlarge the spatial-temporal receptive field. For example, after operating $1^{st}$ tensor separable convolution on the $1^{st}$ sub-dimension, $C_1$ channels interact, and 3D receptive field of such interaction is $3 \times 3 \times 3$. Via further operating $2^{nd}$ tensor separable convolution on the $2^{nd}$ sub-dimension, all $C_1 \times C_2 = C$

channels have feature interaction, and 3D receptive field of such interaction becomes $5 \times 5 \times 5$. This clearly satisfies our principle of feature-interaction sufficiency.

We summarize our contributions in the following. First, we design a novel Channel Tensorized Module (CT-Module), which can achieve convolutional efficiency and feature-interaction sufficiency, via progressively performing spatial/temporal tensor separable convolution along each sub-dimension of the tensorized channel. Second, we equip CT-Module with a distinct Tensor Excitation (TE) mechanism, which can further activate the video features of each sub-operation by spatial, temporal and channel attention in a tensor-wise manner. Subsequently, we apply this full module in a residual block, and flexibly adopt 2D ResNet as our Channel Tensorized Network (CT-Net). In this case, we can gradually enhance feature interaction from a broader 3D receptive field, and learn the key spatial-temporal representation with light computation. Finally, we conduct extensive experiments on a number of popular and challenging benchmarks, e.g., Kinetics (Carreira & Zisserman, 2017), Something-Something V1 and V2 (Goyal et al., 2017b). Our CT-Net outperforms the state-of-the-art methods in terms of classification accuracy and/or computation cost.

## 2 RELATED WORKS

**2D CNN for video classification.** 2D CNN is a straightforward but useful method for video classification (Karpathy et al., 2014; Simonyan & Zisserman, 2014; Wang et al., 2016; Liu et al., 2020; Jiang et al., 2019). For example, Two-stream methods (Simonyan & Zisserman, 2014) learn video representations by fusing the features from RGB and optical flow respectively. Instead of sampling a single RGB frame, TSN (Wang et al., 2016) proposes a sparse temporal sampling strategy to learn video representations. To further improve accuracy, TSM (Lin et al., 2019) proposes a zero-parameter temporal shift module to exchange information with adjacent frames. However, these methods may lack the capacity of learning spatial-temporal interaction comprehensively, which often reduces their discriminative power to recognize complex human actions.

**3D CNN for video classification.** 3D CNN has been widely used to learn a rich spatial-temporal context better (Tran et al., 2015; Carreira & Zisserman, 2017; Feichtenhofer et al., 2019; Sudhakaran et al., 2020; Feichtenhofer, 2020). However, it introduces a lot of parameters, which leads to a difficult optimization problem and large computational load. To resolve this issue, I3D (Carreira & Zisserman, 2017) inflates all the 2D convolution kernels pre-trained on ImageNet, which is helpful for optimizing. Other works also try to factorize 3D convolution kernel to reduce complexity, such as P3D (Qiu et al., 2017) and R(2+1)D (Tran et al., 2018). Recently, CSN (Tran et al., 2019) operates 3D convolution in the depth-wise manner. Nevertheless, all these methods still do not achieve a good trade-off between accuracy and efficiency. To tackle this challenge, we propose CT-Net which learns on spatial-temporal and channel dimensions jointly with lower computation than previous methods.

## 3 METHODS

In this section, we describe our Channel Tensorization Network (CT-Net) in detail. First, we formally introduce our CT-Module in a generic manner. Second, we design a Tensor Excitation (TE) mechanism to enhance CT-Module. Finally, we flexibly adapt ResNet as our CT-Net to achieve a preferable trade-off between accuracy and efficiency for video classification.

### 3.1 CHANNEL TENSORIZATION MODULE

As discussed in the introduction, the previous approaches have problems in convolutional efficiency or feature-interaction sufficiency. To tackle such a problem, we introduce a generic Channel Tensorization Module (CT-Module), by treating the channel dimension of input feature as a multiplication of $K$ sub-dimensions, i.e., $C = C_1 \times C_2 \times \cdots \times C_K$. Naturally, this tensor representation allows to tensorize the kernel size of convolution *TConv()* as a multiplication of $K$ sub-dimensions, too. To simplify the notation, the channel dimension of the output is omitted by default. The output $X_{out}$ can be calculated as follows:

$$\mathbf{X}_{out} = TConv\left(\mathbf{X}_{in}, W^{C_1 \times C_2 \times \cdots \times C_K \times t \times h \times w}\right) \tag{1}$$

where $\mathbf{X}_{in}$ and $W$ denote the tensorized input and kernel respectively. However, such an operation requires large computation, so we introduce the tensor separable convolution to alleviate the issue.

**Tensor Separable Convolution.** We propose to factorize *TConv()* along $K$ channel sub-dimensions. Specifically, we decompose *TConv()* as $K$ tensor separable convolutions *TSConv()*, and apply *TSConv()* sub-dimension by sub-dimension as follows:

$$\mathbf{X}_k = TSConv\left(\mathbf{X}_{k-1}, W^{1 \times \cdots \times C_k \times \cdots \times 1 \times t \times h \times w}\right) \tag{2}$$

where $\mathbf{X}_0 = \mathbf{X}_{in}$ and $\mathbf{X}_{out} = \mathbf{X}_K$. On one hand, the kernel size of the $k^{th}$ *TSConv()* is $(1 \times \cdots \times C_k \times \cdots \times 1 \times t \times h \times w)$. It illustrates that only $C_k$ channels interact in the $k^{th}$ sub-operation, which leads to convolution efficiency. On the other hand, as we stack the *TSConv()*, each convolution performs on the output features of the previous convolution. Therefore, the spatial-temporal receptive field is enlarged. Besides, interactions first occur in $C_1$ channels, second in $C_1 \times C_2$ channels and so on. Finally, $C_1 \times C_2 \times \cdots \times C_K = C$ channels can progressively interact. This clearly satisfies our principle of feature-interaction sufficiency.

**Spatial-Temporal Tensor Separable Convolution.** To further improve convolution efficiency, we factorize the 3D *TSConv()* into 2D spatial *TSConv()* and 1D temporal *TSConv()*. Thus, we can obtain the output features $\mathbf{X}_k^S$ and $\mathbf{X}_k^T$ as follows:

$$\mathbf{X}_k^S = S\text{-}TSConv\left(\mathbf{X}_{k-1}, W^{1 \times \cdots \times C_k \times \cdots \times 1 \times 1 \times h \times w}\right) \tag{3}$$

$$\mathbf{X}_k^T = T\text{-}TSConv\left(\mathbf{X}_{k-1}, W^{1 \times \cdots \times C_k \times \cdots \times 1 \times t \times 1 \times 1}\right) \tag{4}$$

where *S-TSConv()* and *T-TSConv()* represent spatial and temporal tensor separable convolution respectively. Finally, we attempt to aggregate spatial and temporal convolution. There are various connection types of spatial and temporal tensor separable convolution, e.g., parallel and serial types. According to the results of the experiments in Section 4, we utilize the parallel method, which illustrates that we sum the spatial feature $\mathbf{X}_k^S$ and temporal feature $\mathbf{X}_k^T$:

$$\mathbf{X}_k = \mathbf{X}_k^S + \mathbf{X}_k^T \tag{5}$$

## 3.2 TENSOR EXCITATION

Our CT-Module separates feature along spatial, temporal and channel dimensions. To make full use of their cooperative power to learn distinct video features, we design a concise Tensor Excitation (TE) mechanism for each dimension. First of all, we attempt to utilize the TE mechanism to enhance spatial and temporal features respectively. For the spatial feature $\mathbf{X}_k^S$ obtained by Equation 5, our corresponding spatial TE mechanism can be formulated as:

$$\mathbf{U}_k = \mathbf{X}_k^S \otimes Sigmod(S\text{-}TSConv(T\text{-}Pool(\mathbf{X}_k^S))) \tag{6}$$

where *T-Pool()* represents global temporal pooling, i.e., $T \times 1 \times 1$ average pooling. By performing it on $\mathbf{X}_k^S$, we obtain the feature with the size $(C_1 \times C_2 \times \cdots \times C_K \times 1 \times H \times W)$, which gathers spatial contexts along temporal dimension. Subsequently, the spatial tensor separable convolution *S-TSConv()* and the activate function *Sigmod()* are performed to generate the spatial attention heatmap. Finally, the element-wise multiplication $\otimes$ broadcasts the spatial attention along the temporal dimension. Similarly, we perform the temporal TE mechanism for the temporal feature $\mathbf{X}_k^T$:

$$\mathbf{V}_k = \mathbf{X}_k^T \otimes Sigmod(T\text{-}TSConv(S\text{-}Pool(\mathbf{X}_k^T))) \tag{7}$$

where *S-Pool()* and *T-TSConv()* are global spatial pooling and temporal tensor separable convolution correspondingly. At last, after aggregating the spatial and temporal features by addition, i.e., $\mathbf{R}_k = \mathbf{U}_k + \mathbf{V}_k$, we perform a channel-wise TE mechanism as follows:

$$\mathbf{X}_k = \mathbf{R}_k \otimes Sigmod(PW\text{-}TSConv(S\text{-}Pool(\mathbf{R}_k))) \tag{8}$$

We adopt point-wise tensor separable convolution *PW-TSConv()* to learn the weights for aggregating distinctive channels. The rest follows the previous design. Note that all tensor separable convolutions are performed on the same sub-dimension as the previous convolution, which is essentially different from the SE mechanism (Hu et al., 2020). Through the cooperation of the TE mechanism along three different dimensions, the spatial-temporal features can be significantly enhanced.

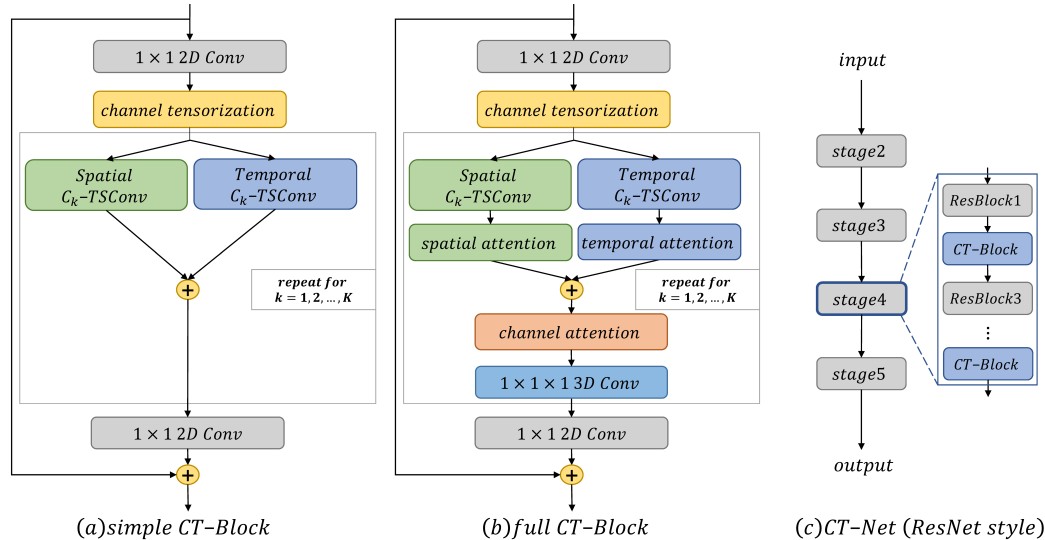

Figure 2: The pipelines of CT-Blocks and the overall architecture of CT-Net. We replace one of every two ResBlocks in ResNet with our CT-Block and the extra point-wise convolution in the last sub-dimension ($k = K$) is ignored. More details can be found in Section3.3.

## 3.3 CHANNEL TENSORIZATIONN NETWORK

We regard ResNet as an exemplar and build up our CT-Net from ResNet50 (or ResNet101). First, we design a simple CT-Block in Figure 2(a), which adapts the $3 \times 3$ convolutional layer in Residual Block (ResBlock) into our CT-Module. It can achieve both convolutional efficiency and feature-interaction sufficiency. Second, we equip our simple CT-Block with the TE mechanism in Figure 2(b), forming a full CT-Block that can improve the cooperative power of all the feature dimensions. Besides, extra point-wise convolutions are added between different sub-operations, which are beneficial for more sufficient feature interaction. At last, we build up a novel CT-Net with CT-Block. As shown in Figure 2(c), we replace one of every two ResBlocks with our CT-Block in every stage. Such a method guarantees a better balance between efficiency and accuracy in our experiments.

**Discussion.** In fact, the popular methods in video classification like C3D, R(2+1)D and CSN (Tran et al., 2015; 2018; 2019) can be viewed as special cases of our CT-Net. We can generate different forms by adjusting three hyper-parameters: **the number of sub-dimensions ($K$), the corresponding dimension size ($C_k$) and the spatial-temporal kernel size ($Kernel_k$)**. To degenerate into C3D, we can set $K = 1$ and $Kernel_1 = 3 \times 3 \times 3$. When $K = 2, Kernel_1 = 1 \times 3 \times 3, Kernel_2 = 3 \times 1 \times 1$ and $C_1 = C_2 = C$ without channel tensorization, it becomes R(2+1)D. Unfortunately, because of lacking the decomposition of channels, C3D and R(2+1)D still have a large computational load. When $K = 2, C = C_1 \times C_2 = C \times 1, Kernel_1 = 1 \times 1 \times 1, Kernel_2 = 3 \times 3 \times 3$, obviously it is equivalent to CSN. However, CSN has a limited receptive field of spatial-temporal interaction. In our CT-Net, we utilize channel tensorization and perform tensor separable convolution along each sub-dimension in turn. Such design can not only preserve interaction among spatial, temporal and channel dimensions but also enlarge the receptive field of feature interaction progressively.

## 4 EXPERIMENTS AND RESULTS

**Datasets and implementation details.** We conduct experiments on three large video benchmarks: Kinetics-400 (Carreira & Zisserman, 2017), Something-Something V1 and V2 (Goyal et al., 2017b). We choose ResNet50 and ResNet101 (He et al., 2016) pre-trained on ImageNet as the backbone and the parameters of CT-Module are randomly initialized. For training, we utilize the dense sampling strategy (Wang et al., 2018) for Kinetics and sparse sampling strategy (Wang et al., 2016) for Something-Something. Random scaling and cropping are applied for data argumentation. Finally, we resize the cropped regions to $256 \times 256$. For testing, we sample multiple clips per video (4 for Kinetics, 2 for others) for pursuing high accuracy, and average all scores for the final prediction.

| Method | 3D Convolution $(t \times h \times w)$ | GFLOPs | Top-1 | Top-5 |
|---|---|---|---|---|
| C3D-Module (Tran et al., 2015) | $Full:\ 3 \times 3 \times 3$ | 59.9 | 46.1 | 75.0 |
| R(2+1)D-Module (Tran et al., 2018) | $Full:\ 1 \times 3 \times 3\ +\ Full:\ 3 \times 1 \times 1$ | 45.8 | 47.0 | 76.1 |
| CSN-Module (Tran et al., 2019) | $Full:\ 1 \times 1 \times 1\ +\ DW:\ 3 \times 3 \times 3$ | 35.6 | 46.8 | 75.7 |
| Our CT-Module | $C_1, \cdots, C_K:\ 1 \times 3 \times 3\ +\ 3 \times 1 \times 1$ | 36.3 | **47.3** | **76.2** |

(a) **Effectiveness of CT-Module.** CT-Module outperforms the recent modules for video modeling.

| Number | GFLOPs | Top-1 | Top-5 |
|---|---|---|---|
| 1D | 45.8 | 46.5 | 75.6 |
| 2D | 36.3 | **47.3** | **76.2** |
| 3D | 35.7 | 47.1 | 75.8 |
| 4D | 35.6 | 46.5 | 75.3 |

(b) **Number of sub-dimensions.** The higher the dimension, the smaller the GFLOPs. 2D channel tensorization achieves the best trade-off.

| Type | Top-1 | Top-5 |
|---|---|---|
| coupling | 47.1 | 76.0 |
| serial | 47.2 | 76.1 |
| parallel | **47.3** | **76.2** |

(c) **Connection type of spatiotemporal convolution.** The parallel connection between spatial and temporal convolution is the best choice.

| $C_2$ | GFLOPs | Top-1 | Top-5 |
|---|---|---|---|
| 1 | 45.9 | 47.0 | 76.1 |
| 4 | 37.6 | 46.8 | 76.0 |
| 16 | 36.4 | 47.2 | 76.0 |
| $\lfloor \sqrt{C} \rfloor$ | 36.3 | **47.3** | **76.2** |

(d) **Dimension size.** $C = C_1 \times C_2$ and the best trade-off is achieved when adopting the rounded middle size $\lfloor \sqrt{C} \rfloor$.

| Number | | GFLOPs | Top-1 | Top-5 |
|---|---|---|---|---|
| +0 | (TSN) | 43.0 | 16.9 | 42.0 |
| +1 | stage5 | 41.9 | 42.3 | 71.1 |
| +4 | stage4-5 | 38.9 | 46.9 | 75.7 |
| +6 | stage3-5 | 37.1 | 47.2 | 76.1 |
| +7 | stage2-5 | 36.3 | **47.3** | **76.2** |
| +12 | stage2-5 | 31.5 | 45.9 | 76.1 |

(e) **Number and location of CT-Blocks.** Simply replacing 1 block in stage5 can bring significant performance improvement. As we replace more blocks from the bottom up, the GFLOPs continues to decrease. Replacing 7 blocks achieves the best trade-off between accuracy and GFLOPs.

| Kernel size | GFLOPs | Top-1 | Top-5 |
|---|---|---|---|
| $C_1\ 1 \times 1 \times 1 \,\|\, 1 \times 1 \times 1$ 
 $C_2\ 1 \times 3 \times 3 \,\|\, 3 \times 1 \times 1$ | 35.5 | 46.1 | 75.1 |
| $C_1\ 1 \times 1 \times 1 \,\|\, 1 \times 1 \times 1$ 
 $C_2\ 1 \times 5 \times 5 \,\|\, 5 \times 1 \times 1$ | 36.6 | 47.2 | 76.2 |
| $C_1\ 1 \times 3 \times 3 \,\|\, 3 \times 1 \times 1$ 
 $C_2\ 1 \times 3 \times 3 \,\|\, 3 \times 1 \times 1$ | 36.3 | 47.3 | 76.2 |
| $C_1\ 1 \times 3 \times 3 \,\|\, 3 \times 1 \times 1$ 
 $C_2\ 1 \times 5 \times 5 \,\|\, 5 \times 1 \times 1$ | 37.4 | 47.5 | 76.4 |
| $C_1\ 1 \times 5 \times 5 \,\|\, 5 \times 1 \times 1$ 
 $C_2\ 1 \times 5 \times 5 \,\|\, 5 \times 1 \times 1$ | 38.9 | 47.6 | 76.5 |

(f) **Kernel sizes along different dimensions.** The larger kernel size brings improvement with more calculation.

| Model | GFLOPs | Top-1 | Top-5 |
|---|---|---|---|
| Baseline (TSN) | 43.0 | 16.9 | 42.0 |
| +CT-Module | 36.3 | 47.3 | 76.2 |
| +CT-Module+PWConv | 37.2 | 48.0 | 76.7 |
| +CT-Module+PWConv+SE | 37.2 | 48.8 | 77.4 |
| +CT-Module+PWConv+TE | 37.3 | **50.1** | **78.8** |

(g) **Impact of different modules.** CT-Module is essential for temporal modeling and TE mechanism is also beneficial.

| Input | | GFLOPs | Top-1 | Top-5 |
|---|---|---|---|---|
| Train: | $224 \times 224$ | 28.6 | 49.1 | 77.4 |
| Test: | $224 \times 224$ | | | |
| Train: | $224 \times 224$ | 37.3 | 49.7 | 77.7 |
| Test: | $256 \times 256$ | | | |
| Train: | $256 \times 256$ | 37.3 | **50.1** | **78.8** |
| Test: | $256 \times 256$ | | | |

(h) **Impact of different spatial resolution.**

Table 2: Ablation studies on Something-Something V1. All models use ResNet50 as the backbone

We follow the same strategy in Non-local (Wang et al., 2018) to pre-process the frames and take 3 crops of $256 \times 256$ as input. Because some multi-clip models in Table 3 and Table 4 sample crops of $256 \times 256$, we simply multiply the GFLOPs reported in the corresponding papers by $(256/224)^2$ for a fair comparison. When considering efficiency, we use just 1 clip per video and the final crop is scaled to $256 \times 256$ to ensure comparable GFLOPs.

## 4.1 ABLATION STUDIES

Table 2 shows our ablation studies on Something-Something V1, which is a challenging dataset that requires video architecture to have a robust spatial-temporal representation ability and is suitable to verify the effectiveness of our method. All models use ResNet50 as the backbone.

**Effectiveness of CT-Module.** In Table 2a, we replace the $3 \times 3$ convolutional layer in ResNet50 with different modules in recent methods (Tran et al., 2015; 2018; 2019). Compared with CSN-Module, our module achieves a better result with similar computation, which reflects the importance of sufficient feature interaction. Besides, it is slightly better than R(2+1)D-Module with much lower calculation, showing the necessity of efficient convolution. Such results demonstrate the effectiveness of our CT-Module. It illustrates our two design principles give preferable guidance for designing an efficient module for temporal modeling.

| Method | Backbone | #Frame | GFLOPs | SomethingV1 | | SomethingV2 | |
| --- | --- | --- | --- | --- | --- | --- | --- |
| | | | | Top-1 | Top-5 | Top-1 | Top-5 |
| ECO$_{EN}$Lite (Zolfaghari et al., 2018) | Incep+3D R18 | 92 | 267 | 46.4 | - | - | - |
| NL I3D + GCN (Wang & Gupta, 2018) | 3D R50 | 32×3×2 | 1818 | 46.1 | 76.8 | - | - |
| ir-CSN (Tran et al., 2019) | 3D R101 | 32×1×10 | 738 | 48.4 | - | - | - |
| ir-CSN (Tran et al., 2019) | 3D R152 | 32×1×10 | 967 | 49.3 | - | - | - |
| CorrNet (Wang et al., 2020) | 3D R50 | 32×1×10 | 1150 | 49.3 | - | - | - |
| TSN (Wang et al., 2016) | 2D R50 | 8 | 33 | 19.7 | 46.6 | 27.8 | 57.6 |
| TSM (Lin et al., 2019) | 2D R50 | 8 | 33 | 45.6 | 74.2 | 59.1 | 85.6 |
| bLVNet-TAM (Fan et al., 2019) | bLR50 | 8×2 | 24 | 46.4 | 76.6 | 59.1 | 86.0 |
| TEINet (Liu et al., 2020) | 2D R50 | 8 | 33 | 47.4 | - | 61.3 | - |
| TEA (Li et al., 2020b) | 2D Res2Net50 | 8 | 35 | 48.9 | 78.1 | - | - |
| PEM+TDLoss (Weng et al., 2020) | 2D R50+TIM | 8 | 33 | 49.8 | - | 62.6 | - |
| PEM+TDLoss (Weng et al., 2020) | 2D R50+TIM | 8×3×2 | 259 | 50.4 | - | 63.5 | - |
| Our CT-Net | 2D R50 | 8 | 37 | 50.1 | 78.8 | 62.5 | 87.7 |
| Our CT-Net | 2D R50 | 8×3×2 | **224** | **51.7** | **80.1** | **63.9** | **88.8** |
| TSN (Wang et al., 2016) | 2D R50 | 16 | 66 | 19.9 | 47.3 | 30.0 | 60.5 |
| TSM (Lin et al., 2019) | 2D R50 | 16 | 66 | 47.2 | 77.1 | 63.4 | 88.5 |
| bLVNet-TAM (Fan et al., 2019) | bLR50 | 16×2 | 48 | 48.4 | 78.8 | 61.7 | 88.1 |
| TEINet (Liu et al., 2020) | 2D R50 | 16 | 66 | 49.9 | - | 62.1 | - |
| TEA (Li et al., 2020b) | 2D Res2Net50 | 16 | 70 | 51.9 | 80.3 | - | - |
| PEM+TDLoss (Weng et al., 2020) | 2D R50+TIM | 16 | 66 | 50.9 | - | 63.8 | - |
| PEM+TDLoss (Weng et al., 2020) | 2D R50+TIM | 16×3×2 | 517 | 52.0 | - | 65.0 | - |
| Our CT-Net | 2D R50 | 16 | 75 | 52.5 | 80.9 | 64.5 | 89.3 |
| Our CT-Net | 2D R50 | 16×3×2 | **447** | **53.4** | **81.7** | **65.9** | **90.1** |
| Our CT-Net$_{EN}$ | 2D (R50)×4 | 8+12+16+24 | **280** | **56.6** | **83.9** | **67.8** | **91.1** |

Table 3: **Comparison with the state-of-the-art on Something-Something V1&V2.** Our CT-Net$_{16f}$ outperforms all the single-clip models in Something-Something and even better than most of the multi-clip models. And our CT-Net$_{EN}$ outperforms all methods with very lower computation.

**Number of sub-dimensions.** Increasing the number of sub-dimensions saves a lot of computation, but the corresponding accuracy first increases and then decreases as shown in Table 2b. Compared with the 1D method, the 4D method significantly reduces GFLOPs, achieving comparable accuracy. As for the decrease of accuracy when $K$ is too large, we argue that the number of channel in the shallow layer is small (64/128), thus there are too few channels in a single dimension, leading to insufficient feature-interaction. Since the 2D method obtains the best trade-off, we set $K = 2$ in all the following experiments.

**Connection type of spatiotemporal convolution.** The coupling $3 \times 3 \times 3$ convolution can be decomposed into serial or parallel spatial/temporal convolution. Table 2c reveals that factorizing the 3D kernel can boost results as expected. Besides, the parallel connection is better, thus we adopt parallel connection as the default.

**Dimension size.** As we set $K = 2$, it is essential to explore the impact of changing the dimension size $C_2$. We can demonstrate that the computation is the lowest when $C_1 = C_2 = \sqrt{C}$. Since $C$ is not always a perfect square number, we adopt the rounded middle size $\lfloor \sqrt{C} \rfloor$. Table 2d shows that when $C_2 = \lfloor \sqrt{C} \rfloor$, the model not only requires the fewest computation cost but also achieves the best performance. Hence, we set $C_2 = \lfloor \sqrt{C} \rfloor$ naturally.

**Number and location of CT-Blocks.** Table 2e illustrates that simply replacing 1 block in stage5 can bring significant performance improvement (16.9% vs. 42.3%). As we replace more blocks from the bottom up, the GFLOPs continues to decrease. Moreover, the bottom blocks seem to be more beneficial to temporal modeling, since replacing the extra 3 blocks in stage2 and stage3 only improve the accuracy by 0.4% (46.9% vs. 47.3%). Since replacing 7 blocks achieves the highest accuracy, we replace 7 blocks by default.

**Kernel sizes along different dimensions.** In Table 2f, we can observe that two concatenated $3^3$ convolution kernels are slightly better than those with the same receptive field ($1^3+5^3$). Furthermore, the larger kernel size can bring performance improvement but more calculation. It reveals that our CT-Module avoids the limited receptive field of feature interaction, and it can progressively enlarge the receptive field of such interaction on all the dimensions. Considering a better trade-off between accuracy and computation, we choose two concatenated $3^3$ convolution kernels.

| Method | Backbone | #Frame | GFLOPs | Top-1 | Top-5 |
|---|---|---|---|---|---|
| R(2+1)D (Tran et al., 2018) | 2D R34 | 32×1×10 | 1520=152×10 | 72.0 | 91.4 |
| TSN (Wang et al., 2016) | Inception | 25×10×1 | 800=80×10 | 72.5 | 90.2 |
| I3D (Carreira & Zisserman, 2017) | Inception | 64×N/A×N/A | 108×N/A | 71.1 | 89.3 |
| TSM (Lin et al., 2019) | 2D R50 | 16×3×10 | 2580=86×30 | 74.7 | - |
| TEINet (Liu et al., 2020) | 2D R50 | 16×3×10 | 2580=86×30 | 76.2 | 92.5 |
| bLVNet-TAM (Fan et al., 2019) | bLR50 | (16×2)×3×3 | 561=62.3×9 | 72.0 | 90.6 |
| TEA (Li et al., 2020b) | 2D Res2Net50 | 16×3×10 | 2730=91×30 | 76.1 | 92.5 |
| PEM+TDLoss (Weng et al., 2020) | 2D R50+TIM | 16×3×10 | 2580=86×30 | 76.9 | 93.0 |
| CorrNet (Wang et al., 2020) | 3D R50 | 32×1×10 | 1150=115×10 | 77.2 | - |
| SlowFast (Feichtenhofer et al., 2019) | 3D R50+R50 | 36=(4+32)×3×10 | 1083=36.1×30 | 75.2 | 91.5 |
| SlowFast (Feichtenhofer et al., 2019) | 3D R50+R50 | 40=(8+32)×3×10 | 1971=65.7×30 | 76.4 | 92.2 |
| Our CT-Net | 2D R50 | 16×3×4 | **895=74.6×12** | **77.3** | **92.7** |
| X3D-XL (Feichtenhofer, 2020) | - | 16×3×10 | 1452=48.4×30 | 79.1 | 93.9 |
| SmallBigNet (Li et al., 2020a) | 2D R101 | 32×3×4 | 6552=546×12 | 77.4 | 93.3 |
| ip-CSN (Tran et al., 2019) | 3D R101 | 32×3×10 | 2490=83.0×30 | 76.8 | 92.5 |
| ip-CSN (Tran et al., 2019) | 3D R152 | 32×3×10 | 3264=108.8×30 | 77.8 | 92.8 |
| CorrNet (Wang et al., 2020) | 3D R101 | 32×3×10 | 6720=224×30 | 79.2 | - |
| SlowFast (Feichtenhofer et al., 2019) | 3D R101+R101 | 40=(8+32)×3×10 | 3180=106×30 | 77.9 | 93.2 |
| SlowFast (Feichtenhofer et al., 2019) | 3D R101+R101 | 80=(16+64)×3×10 | 6390=213×30 | 78.9 | 93.5 |
| NL I3D (Wang & Gupta, 2018) | 3D R101 | 128×3×10 | 10770=359×30 | 77.7 | 93.3 |
| Our CT-Net | 2D R101 | 16×3×4 | 1746=145.5×12 | 78.8 | 93.7 |
| Our CT-Net$_{EN}$ | 2D R50+R101 | (16+16)×3×4 | **2641=220.1×12** | **79.8** | **94.2** |

Table 4: **Comparison with the state-of-the-art on Kinetics-400.** It shows that CT-Net-R50$_{16f}$ can surpass all existing lightweight models and even SlowFast-R50$_{40f}$. When fusing different models, our model is $2.4\times$ faster than SlowFast-R101$_{80f}$ and shows an $0.9\%$ performance gain.

**Impact of different modules and different spatial resolution.** In Table 2g, our CT-Module can significantly boost its baseline (16.9% vs. 47.3%) and the TE mechanism can further improve the accuracy by 2.1% (48.0% vs. 50.1%). The extra point-wise convolution also boosts performance, which demonstrates that it is beneficial for sufficient feature interaction. Compared with the SE mechanisms, our TE mechanism focuses more on features in different sub-dimensions individually, thus effectively enhancing spatial-temporal features. In our experiments, to ensure GFLOPs is comparable with other methods, we crop the input to $256 \times 256$ during testing. Table 2h shows both training and testing with a larger spatial resolution of input bring clear performance improvement.

## 4.2 COMPARISONS WITH THE STATE-OF-THE-ARTS

**Something-Something V1&V2.** We make a comprehensive comparison in Table 3. Compared with NL I3D+GCN$_{32f}$, our CT-Net$_{8f}$ gains 4.0% top-1 accuracy improvement with $49.1\times$ fewer GFLOPs in Something-Something V1. Besides, our CT-Net$_{8f}$ (51.7%) is better than the ir-CSN$_{32f}$ (49.3%) which adopts ResNet-152 as the backbone. Moreover, our CT-Net$_{16f}$ outperforms all the single-clip models in Something-Something V1&V2 and even better than most of the multi-clip models. It illustrates that our CT-Net is preferable to capture temporal contextual information efficiently. Surprisingly, with only 280 GFLOPs, our ensemble model CT-Net$_{EN}$ achieves 56.6%(67.8%) top-1 accuracy in Something-Something V1(V2), which outperforms all methods.

**Kinetics-400.** Kinetics-400 is a large-scale sence-related dataset, and the lightweight 2D models are usually inferior to the 3D models on it. Table 4 shows our CT-Net-R50$_{16f}$ can surpass all existing lightweight models based on 2D backbone. Even compared with SlowFast-R50$_{40f}$, our CT-Net-R50$_{16f}$ also achieves higher accuracy (77.3% vs. 76.4%). Note that our reproduced SlowFast-R50 performs worse than that in the paper (Feichtenhofer et al., 2019), which may result from the missing videos in Kinetics-400. As for the deeper model, compared with SlowFast-R101$_{80f}$, our CT-Net-R101$_{16f}$ requires $3.7\times$ fewer GFLOPs but gains comparable results (78.8% vs. 78.9%). Besides, it achieves comparable top-1 accuracy with X3D-XL (78.8% vs. 79.1%) under a similar GFLOPs. However, X3D requires extensive model searching with an expensive GPU setting, while our CT-Net can be trained traditionally with feasible computation. We perform score fusion over CT-Net-R50$_{16f}$ and CT-Net-R101$_{16f}$, which mimics two-steam fusion with two temporal rates. In this setting, our model is $2.4\times$ faster than SlowFast-R101$_{80f}$ and shows an $0.9\%$ performance gain (79.8% vs. 78.9%) but only uses 32 frames.

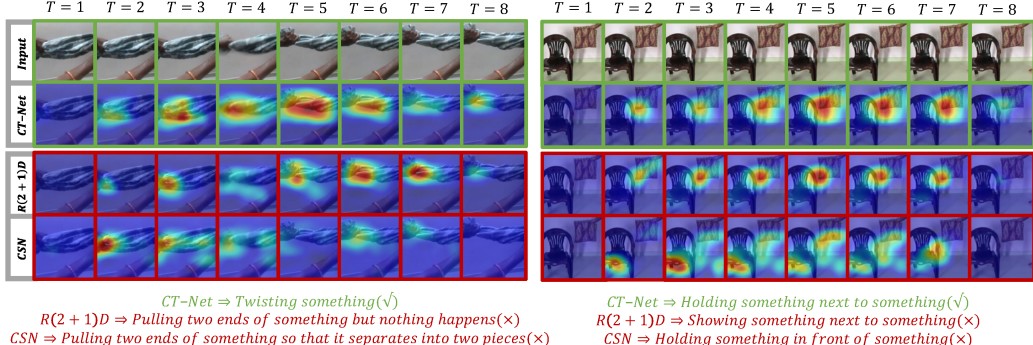

Figure 3: **Comparison of visualization.** Videos are sampled from Something-Something V1. Compared with R(2+1)D and CSN, our CT-Net can localize the action and object better both in space and time thanks to the larger spatial-temporal receptive field.

### 4.3 VISUALIZATION

We use Saliency Tubes (Stergiou et al., 2019) to generate the visualization, for it can show the most discriminative features that the network locates. In Figure 3, we sample two videos from Something-Something V1 which requires complex temporal modeling. In the left example, our CT-Net focuses on a larger area around the towel, especially in the fourth and fifth frames, thus predicting that someone is twisting it. In contrast, R(2+1)D only concentrates on one side of the towel and gives the wrong judgment. The same situation can be seen in the right example. We argue that CT-Net can localize the action and object accurately thanks to the larger spatial-temporal receptive field. As for CSN, the regions of interest seem to be scattered, because it lacks sufficient spatial-temporal interaction, thus ignoring the rich context both in space and time.

## 5 CONCLUSIONS

In this paper, we construct an efficient tensor separable convolution to learn the discriminative video representation. We view the channel dimension of the input feature as a multiplication of K sub-dimensions and stack spatial/temporal tensor separable convolution along each of K sub-dimensions. Moreover, CT-Module is cooperated with the Tensor Excitation mechanism to further improve performance. All experiments demonstrate that our concise and novel CT-Net obtains a preferable balance between accuracy and efficiency on large-scale video datasets. Our proposed principles are preferable guidance for designing an efficient module for temporal modeling.

## 6 ACKNOWLEDGEMENT

This work is partially supported by National Natural Science Foundation of China (6187617, U1713208), the National Key Research and Development Program of China (No. 2020YFC2004800), Science and Technology Service Network Initiative of Chinese Academy of Sciences (KFJ-STS-QYZX-092), Shenzhen Institute of Artificial Intelligence and Robotics for Society.

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

## A  Appendix

### A.1  More training details

We use SGD with momentum 0.9 and cosine learning rate schedule (Loshchilov & Hutter, 2017) to train the entire network. The first 10 epochs are used for warm-up (Goyal et al., 2017a) to overcome early optimization difficulty. For kinetics, the batch, total epochs, initial learning rate, dropout and weight decay are set to 64, 110, 0.01, 0.5 and 1e-4 respectively. All these hyper-parameters are set to 64, 45, 0.02, 0.3 and 5e-4 respectively for Something-Something.

### A.2  Tensor Excitation Mechanism

The implementation of our Tensor Excitation is shown in Figure 4. Different from the SE module, we use tensor separable convolution in the TE mechanism. Moreover, when obtaining the spatial attention, we squeeze the temporal dimension and perform spatial tensor separable convolution, because temporal information is insignificant for spatial attention and vice versa. We add the Batch Normalization (BN) layer for better optimization.

### A.3  Results on UCF101 and HMDB51

To verify the generation ability of our CT-Net on smaller datasets, we conduct transfer learning experiments from Kinetics400 to UCF101 (Soomro et al., 2012) and HMDB-51 (Kuehne et al., 2011). We test CT-Net with 16 input frames and evaluate it over three splits and report the averaged results. As shown in Table 5, our CT-Net$_{16f}$ achieves competitive performance when compared with the recent methods, which demonstrates the generation ability of our CT-Net.

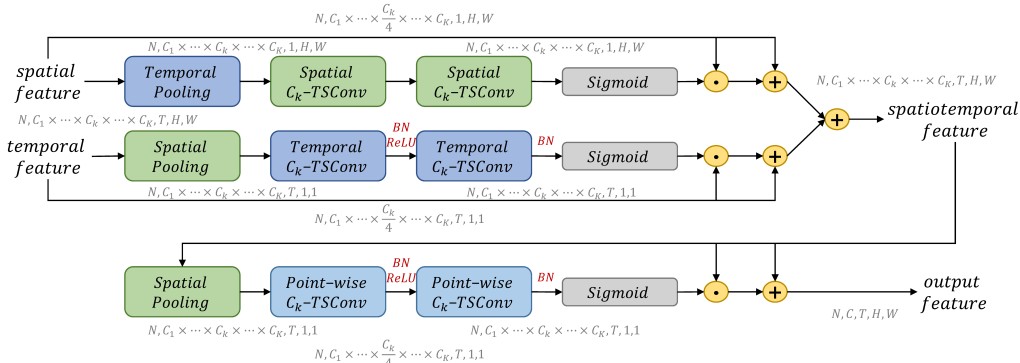

Figure 4: The implementation of our Tensor Excitation (TE) mechanism.

| Method | Backbone | Pretrain | UCF101 | HMDB51 |
|---|---|---|---|---|
| C3D(Tran et al., 2015) | 3D VGG-11 | Sports-1M | 82.3 | 51.6 |
| I3D(Carreira & Zisserman, 2017) | 3D Inception | ImageNet+Kinetics | 95.1 | 74.3 |
| ECO(Zolfaghari et al., 2018) | Inception+3D R18 | ImageNet+Kinetics | 94.8 | 72.4 |
| TSN(Wang et al., 2016) | Inception | ImageNet+Kinetics | 91.1 | - |
| TSM(Lin et al., 2019) | 2D R50 | ImageNet+Kinetics | 94.5 | 70.7 |
| STM(Jiang et al., 2019) | 2D R50 | ImageNet+Kinetics | 96.2 | 72.2 |
| Our CT-Net | 2D R50 | ImageNet+Kinetics | 96.2 | 73.2 |

Table 5: Comparison results on UCF101 and HMDB51.

## A.4 MORE RESULTS ON SOMETHING-SOMETHING V1&V2

Table 6 shows more results on Something-Something V1&V2. We train CT-Net with a different number of input frames and then test these models with different sampling strategies. We average the prediction scores obtained from the previous models to evaluate the ensemble models. With more input frames, the corresponding accuracy becomes higher. As for the reason that CT-Net$_{24f}$ is worse than CT-Net$_{16f}$, we argue that is because the model is hard to optimize with too many input frames. Sampling more clips or more crops also boosts performance. Moreover, our ensemble models gain the state-of-the-art top-1 accuracy of 56.6%(68.3%) on Something-Something V1(V2).

## A.5 MORE ABLATION STUDIES ON MINI-KINETICS AND SOMETHING-SOMETHING V2

To verify the effectiveness of our module comprehensively, we also conduct experiments in Mini-Kinetics and Something-Something V2 and report the multi-clip accuracy and single-clip accuracy respectively. Mini-Kinetics covers 200 action classes and is a subset of Kinetics-400, while Something-Something V2 covers the same action classes as Something-Something V1 but contains more videos. As shown in Table 7, the performance trend of different modules is similar to that shown in Table 2a. Since Mini-Kinetics does not highly depend on temporal modeling, the gap becomes smaller but still demonstrates the effectiveness of our CT-Module.

## A.6 ADAPTING DIFFERENT PRE-TRAINED IMAGENET ARCHITECTURES AS CT-NET

In fact, by directly replacing the 3×3 convolution with our CT-Module, we can easily adapt different pre-trained ImageNet architectures as CT-Net. Table 8 shows that it is also sensible to use InceptionV3 as the backbone. We believe that through more elaborate design, our CT-Net based on different backbones can achieve comparable performance.

## A.7 VALIDATION PLOT

In Figure 5, we plot the accuracy vs per-clip GFLOPs on Kinetics-400. It reveals that our CT-Net achieves a better trade-off than most of the existing methods on Kinetics-400.

| Method | #Frame | GFLOPs | #Param | SomethingV1 Top-1 | Top-5 | SomethingV2 Top-1 | Top-5 |
|---|---|---|---|---|---|---|---|
| Our CT-Net | 8 | 37 | | 50.1 | 78.8 | 62.5 | 87.7 |
| | 12 | 56 | 21.0M | 52.1 | 80.0 | 63.9 | 88.7 |
| | 16 | 75 | | 52.5 | 80.9 | 64.5 | 89.3 |
| | 24 | 112 | | 52.5 | 80.9 | 64.6 | 89.1 |
| | $8\times1\times2$ | 75 | | 51.6 | 79.7 | 63.5 | 88.5 |
| | $12\times1\times2$ | 112 | 21.0M | 52.8 | 80.6 | 64.6 | 89.3 |
| | $16\times1\times2$ | 151 | | 53.2 | 81.3 | 65.2 | 89.7 |
| | $24\times1\times2$ | 224 | | 52.9 | 81.3 | 65.0 | 89.3 |
| | $8\times3\times2$ | 224 | | 51.7 | 80.1 | 63.9 | 88.8 |
| | $12\times3\times2$ | 336 | 21.0M | 53.0 | 81.1 | 65.3 | 89.6 |
| | $16\times3\times2$ | 447 | | 53.4 | 81.7 | 65.9 | 90.1 |
| | $24\times3\times2$ | 672 | | 53.6 | 81.6 | 65.5 | 89.8 |
| Our CT-Net$_{EN}$ | $8+16$ | 112 | | 54.4 | 82.0 | 66.2 | 90.4 |
| | $(8+12+16+24)\times1\times2$ | 280 | 83.8M | 56.6 | 83.9 | 67.8 | 91.1 |
| | $(8+12+16+24)\times1\times2$ | 560 | | 56.6 | 84.0 | 67.8 | 91.3 |
| | $(8+12+16+24)\times3\times2$ | 1679 | | 56.6 | 83.9 | 68.3 | 91.3 |

Table 6: More results on Something-Something V1&V2.

| Method | Backbone | GFLOPs | Mini-Kinetics Top-1 | Top-5 | SomethingV2 Top-1 | Top-5 |
|---|---|---|---|---|---|---|
| C3D-Module (Tran et al., 2015) | 2D R50 | 59.9 | 77.5 | 93.0 | 59.1 | 85.5 |
| R(2+1)D-Module (Tran et al., 2018) | 2D R50 | 45.8 | 77.8 | 93.2 | 60.0 | 86.0 |
| CSN-Module (Tran et al., 2019) | 2D R50 | 35.6 | 77.6 | 93.2 | 59.5 | 86.0 |
| Our CT-Module | 2D R50 | 36.3 | **78.0** | **93.6** | **60.3** | **86.4** |

Table 7: More ablation studies on Mini-Kinetics and Something-Something V2.

| Method | Backbone | GFLOPs | #Param.(M) | Top-1 | Top-5 |
|---|---|---|---|---|---|
| Baseline (TSN) | 2D ResNet-50 | 43.0 | 23.9 | 16.9 | 42.0 |
| Our CT-Net | 2D ResNet-50 | 37.3 | 21.0 | **50.1** | **78.8** |
| Baseline (TSN) | InceptionV3 | 45.8 | 22.1 | 18.3 | 43.9 |
| Our CT-Net | InceptionV3 | 43.9 | 20.9 | 47.2 | 76.1 |

Table 8: Adapting different pre-trained ImageNet architectures as CT-Net.

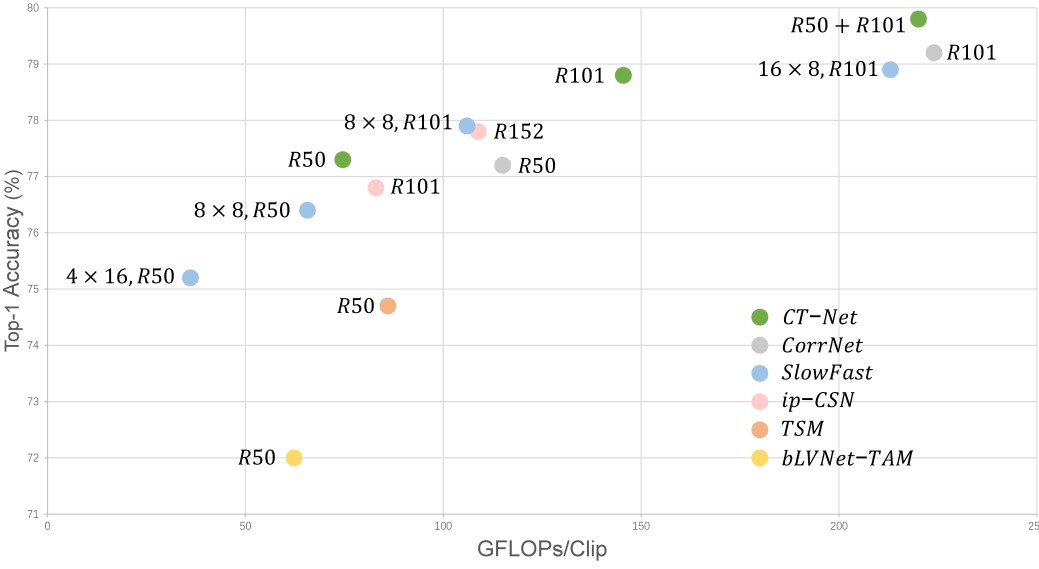

Figure 5: Accuracy vs per-clip GFLOPs on Kinetics-400.