# OpenReview forum: "CT-Net: Channel Tensorization Network for Video Classification"
_ICLR.cc/2021/Conference — ICLR 2021 Poster_

### Official Review · AnonReviewer2 · 2020-10-28
**Official Blind Review#2**

**Rating:** 7
**Confidence:** 3

**Review:**

The paper proposes a novel Channel Tensorized Module (CT-Module) to construct an efficient tensor separable convolution and learn the discriminative video representation. The proposed solution achieves a preferable balance between convolutional efficiency and feature-interaction sufficiency. The experiments were conducted on 3 benchmark video-classification datasets outperforming the state-of-the-art results.

Strengths:
- The paper's novelty is the first method for exploring the spatial/temporal tensor separable convolution along each sub-dimension.
- The CT-Network framework is rigorously presented.
- Experimental results outperforming state-of-the-art results.
- The Paper is well written and easy to follow.

Weaknesses:
- The abstract is too long.
- Results of the Sport1M dataset (standard video classification dataset) are missing.
- Table 2 looks clumsy and hard to understand the results of sub-tables.
- What are the reasons for choosing something-something V1 dataset for ablation studies? How about the ablation experiments on Kinetic and something-something V2 dataset?.
- Since earlier modes were having better performance on the Resnet-152 model, the authors could have experimented with the Resent-152 model as well. Why only 50 and 101?.
- This paper needs careful proof-reading as there are major typos across the paper.


Major Comments:
- Why do authors adapt ResNet as proposed CT-Net? Any reason for this?
- Can we adapt other pre-trained Imagenet architectures as CT-Net?
- The dataset used for the comparison of visualization in Fig.3 is missing.
- If authors can showcase the performance of proposed models on other domain datasets (say 4D fMRI or 3D medical imaging datasets), it will be more interesting.
- validation plots are missing: the accuracy vs channel-interaction plot and the accuracy vs G-Flops per clip.

Typos:
-  Page7:  In our experiments, to ensure GFLOPs is comparable with other methods. we crop the input to 256 × 256 during testing. (Sentence continuity is missing, and check “we”).

Addressed Concerns:
- corrected the typos.
- Majority of the weaknesses are addressed.

---

> ### Author Response · Authors · 2020-11-23
> **Response to Reviewer 2 (Part 1)**
>
> Thanks for your constructive comments. We provide our feedbacks and modify our paper as follows:
>
> ------
>
> **Q1:  The abstract is too long /Table 2 looks clumsy/Need careful proof-reading.**
>
> **A1:** Thanks for this comment. We have simplified them (abstract on page 1 and Table 2 on page 6) and corrected the typos in the updated paper.
>
> ------
>
> **Q2:  Results of the Sport1M dataset are missing.**
>
> **A2:** There are two main reasons.
>
> First, in the last two years of the action recognition community, most approaches (such as Non-local [1], SlowFast [2], X3D [3] and TSM [4]) are evaluated on Kinetics400 and/or Something-Something. Hence, we follow them to make a fair and rigorous comparison.
>
> Second, Sports1M has been collected before 2014, from public videos on Youtube. The links of many videos are actually no longer valid, which makes experimental comparison infeasible. Thanks for this suggestion, and we will take it as our future work.
>
> ------
>
> **Q3: What are the reasons for choosing something-something V1 dataset for ablation studies? How about the ablation experiments on Kinetic and something-something V2 dataset?**
>
> **A3:** There are two main reasons.
>
> First, Something-Something V1 is a challenging benchmark that exhibits complex spatial-temporal action evolutions in videos. Hence, it is a suitable dataset to evaluate video feature interactions of our CT-Net.
>
> Second, Something-Something V1 is relatively smaller than Kinetics and Something-Something V2. Using this dataset for ablation studies allows us to investigate all the detailed designs of CT-Net, with feasible computation cost.
>
> Due to the time limitation of rebuttal, we further conduct some ablation studies on Mini-Kinetics (a subset of Kinetics400) and Something-Something V2, and report the results in Table 7 (on page 12) of our updated appendix. As expected, the performance trend is similar to that on Something-Something V1 (i.e., Table 2 (a)).
>
> | Model          | Backbone | GFLOPs | Mini-Kinetics Top-1&Top5 | Something-Something V2 Top-1&Top5 |
> | :------------- | :-------: | :----: | :-----------------------: | :--------------------------------: |
> | C3D-Module     |   2D R50 |  59.9  |        77.5 & 93.0        |            59.1 & 85.5             |
> | R(2+1)D-Module |   2D R50 |  45.8  |        77.8 & 93.2        |            60.0 & 86.0             |
> | CSN-Module     |   2D R50 |  35.6  |        77.6 & 93.2        |            59.5 & 86.0             |
> | Our CT-Module  |   2D R50 |  36.3  |    **78.0** & **93.6**    |        **60.3** & **86.4**         |
>
> ------
>
> **Q4: Why to choose ResNet-50 and ResNet-101? How about ResNet-152？**
>
> **A4:** There are three main reasons.
>
> First, for efficient video classification, most existing approaches (such as TSM [4], STM [5], TEA [6] and GST [7]) adopt ResNet-50 and ResNet-101 as the backbone. We follow them for a fair comparison.
>
> Second, as shown in Table 3, our CT-Net-50 achieves even better accuracy than ir-CSN-152 on Something- Something V1, showing the power of our method.
>
> Third, although deeper network would perform better, this brings expensive computation cost which is infeasible for our current computation resource. We will take it as our future work.
>
> ------
>
> **Q5: (1) Why do authors adapt ResNet as proposed CT-Net? (2) Can we adapt other pre-trained Imagenet architectures as CT-Net?**
>
> **A5:** (1) ResNet is a widely-used architecture with standard and effective residual blocks. Most recent works on video classification (e.g. Non-local [1], SlowFast [2], TSM [4] and TEA [6]) choose it as the backbone to show effectiveness. Hence, we choose it for a fair comparison with other methods.
>
> (2) It is possible to adapt other networks as CT-Net. We adapt InceptionV3 as our CT-Net. As shown in Table 8 (on page 13) of our updated appendix, CT-Net with InceptionV3 as backbone also works well.
>
> | Method         |   Backbone   | GFLOPs | **\#Param.(M)** |  Top-1   |  Top-1   |
> | :------------- | :----------: | :----: | :-------------: | :------: | :------: |
> | Baseline (TSN) | 2D ResNet-50 |  43.0  |      23.9       |   16.9   |   42.0   |
> | Our CT-Net     | 2D ResNet-50 |  37.3  |      21.0       | **50.1** | **78.8** |
> | Baseline (TSN) | InceptionV3  |  45.8  |      22.1       |   18.3   |   43.9   |
> | Our CT-Net     | InceptionV3  |  43.9  |      20.9       |   47.2   |   76.1   |
>
> ------

---

> > ### Author Response · Authors · 2020-11-23
> > **Response to Reviewer 2 (Part 2)**
> >
> > **Q6: The dataset used for the comparison of visualization in Fig.3 is missing.**
> >
> > **A6:** It is Something-Something V1. We have actually mentioned it in Section 4.3. For clarity, we have added a detailed description in the caption of Fig.3  (on page 8).
> >
> > ------
> >
> > **Q7: If authors can showcase the performance of proposed models on other domain datasets (say 4D fMRI or 3D medical imaging datasets), it will be more interesting.**
> >
> > **A7:** Channel Tensorization (CT) is mainly designed to enhance spatial-temporal and channel interactions for human action recognition in videos. Investigating it for other areas is beyond our goal in this paper. Thanks for your suggestion, and we will take it as our future work.
> >
> > ------
> >
> > **Q8: Validation plots are missing: the accuracy vs channel-interaction plot and the accuracy vs G-Flops per clip.**
> >
> > **A8:** In fact, these two plots have been demonstrated in Table 2(e). As we increase the number of CT blocks, feature interactions are increasing with convolution efficiency. As expected, the accuracy is gradually improved, and GFLOPs is getting smaller. Additionally, accuracy vs. per-clip GFLOPs has also shown in Table 3 and Table 4. For better comparing SOTA approaches, we have added the plot of the accuracy vs. per-clip GFLOPs to the appendix  (Figure 5 on page 13), which reveals our CT-Net achieves a better trade-off.
> >
> > ------
> >
> > [1] Wang X, Girshick R, Gupta A, et al. Non-local neural networks[C]//Proceedings of the IEEE conference on computer vision and pattern recognition. 2018: 7794-7803.
> >
> > [2] Feichtenhofer C, Fan H, Malik J, et al. Slowfast networks for video recognition[C]//Proceedings of the IEEE international conference on computer vision. 2019: 6202-6211.
> >
> > [3] Feichtenhofer C. X3D: Expanding Architectures for Efficient Video Recognition[C]//Proceedings of the IEEE/CVF Conference on Computer Vision and Pattern Recognition. 2020: 203-213.
> >
> > [4] Lin J, Gan C, Han S. Tsm: Temporal shift module for efficient video understanding[C]//Proceedings of the IEEE International Conference on Computer Vision. 2019: 7083-7093.
> >
> > [5] Jiang B, Wang M M, Gan W, et al. Stm: Spatiotemporal and motion encoding for action recognition[C]//Proceedings of the IEEE International Conference on Computer Vision. 2019: 2000-2009.
> >
> > [6] Li Y, Ji B, Shi X, et al. TEA: Temporal Excitation and Aggregation for Action Recognition[C]//Proceedings of the IEEE/CVF Conference on Computer Vision and Pattern Recognition. 2020: 909-918.
> >
> > [7] Luo C, Yuille A L. Grouped spatial-temporal aggregation for efficient action recognition[C]//Proceedings of the IEEE International Conference on Computer Vision. 2019: 5512-5521.

---

### Official Review · AnonReviewer3 · 2020-10-28
**CT-Net**

**Rating:** 7
**Confidence:** 4

**Review:**

Summary

The paper proposes a new architecture for lightweight action classification networks, named Channel Tensorization Network (CT-Net). The idea of this architecture is the tensorization of mid-level input features in combination with an attention mechanism that allows to select relevant features. the channel tensorization can be used as intermediate building block e.g. in a ResNet alternately with a Res block.
The ablation study is done on Something-Something V1 and the overall method is compared to state-of-the-art on Something-Something V1&V2 and Kinetics-400 and is able to outperform other lightweight architecture with comparable size.


Paper strengths

- I think the ideas proposed in this paper are interesting and I would not be aware of any architecture that would use a similar arrangement of tensorization and attention in the mid-layer part to allow for a lightweight 3D convolution architecture.

- The paper shows competitive results on Something-Something as well as on Kinetics-400

- The visualization supports the claim that the attention mechanism seems to learn to focus more on relevant parts of the video clip.


Paper weakness

- The paper does not introduce a new technique per se, but manages to combine known elements. So there might be the issue of limited novelty.

Conclusion

I think that the overall idea of reducing the computational load in 3d convolution architectures is a valid and important topic, as simply increasing networks and training data is not working for the majority of tasks. Therefore, lightweight but powerful architectures have a need in the field. I understand that the proposed elements have only a limited novelty, but I think that the clever integration in this case might justify a publication in this case.

---

> ### Author Response · Authors · 2020-11-23
> **Response to Reviewer 3**
>
> **Q1:  The paper does not introduce a new technique per se, but manages to combine known elements. So there might be the issue of limited novelty.**
>
> **A1:** First, thanks for your positive review overall. Here we would like to further emphasize our novelty.
>
> (1) The most important contribution is the channel tensorization. Via such high-dimension design, we can achieve convolution efficiency and feature interaction sufficiency on all spatial, temporal and channel dimensions. This has not been fully explored by the previous works, to our best knowledge.
>
> (2) Different from attention in the literature, our tensor attention is performed on all spatial, temporal and channel dimensions of video features. As a result, it can cooperatively highlight action from different dimensions to further enhance the performance of the CT-Net.
>
> (3) Our CT-Net outperforms a number of recent SOTA approaches in Kinetics-400 and Something-Something V1&V2, in terms of accuracy and/or efficiency.

---

### Official Review · AnonReviewer4 · 2020-10-29
**An interesting idea that probably needs more justification.**

**Rating:** 5
**Confidence:** 2

**Review:**

This manuscript proposes a novel convolutional operation for learning representations from video data. By decomposing the channel dimension into sub dimensions in the typically 4D video data (Time, Channel, Width, Height), one defines spatial-temporal separable convolution for each sub-dimension. This could improve the representation learning in term of efficiency and modeling quality.

Pro: the experiments seem quite thorough.

Con: i) I'm not fully convinced that it is beneficial to decompose the channel dimension, which is usually only 3 and relatively small compared with the other dimensions. Would it be possible / sensible to apply tensorization on the time dimension? ii) The ablation study could have included the comparison between only channel tensorization and channel tensorization and self attention, which would highlight the new contribution of the paper.

I'd be happy to update my rating if the authors would elaborate these two points.

---

> ### Author Response · Authors · 2020-11-23
> **Response to Reviewer 4**
>
> Thanks for your constructive comments.
>
> ------
>
> **Q1: (1) I’m not fully convinced that it is beneficial to decompose the channel dimension, which is usually only 3 and relatively small compared with the other dimensions. (2) Would it be possible or sensible to apply tensorization on the time dimension?**
>
> **A1:** (1) Actually, we decompose channel dimensions of the CNN feature, instead of the input image. Taking ResNet as an example, channel dimensions in stage 2/3/4/5 are 64/128/256/512 respectively, which are large enough to apply our tensorization.
>
> (2) We agree that it is possible to apply tensorization on the time dimension. However, it is more sensible to perform tensorization on the channel dimension, since the time dimension is much smaller than the channel dimension, e.g., the popular choice of time dimension is from 8 to 32 sampled frames. Following your suggestion, we did tensorization on time dimension (8 sampled frames). The top-1 accuracy on Something-Something V1 is 44.0%, which is worse than our CT-Net (47.3%) as expected.
>
> ------
>
> **Q2: The ablation study could have included the comparison between only channel tensorization and channel tensorization and self attention, which would highlight the new contribution of the paper.**
>
> **A2:** We have actually done this comparison in Table 2(g). Adding channel tensorization can achieve 30.4% accuracy improvement (baseline vs. baseline+CT: 16.9% vs 47.3% on Something-Something V1). Adding attention would further achieve 2.1% accuracy improvement (48.0% vs. 50.1%). Clearly, the CT module makes the most important contribution to our design.
>
> | Model                | GFLOPs |    Top-1 |    Top-5 |
> | -------------------- | :-----: | :-------: | :-------: |
> | Baseline (TSN)       |   43.0 |     16.9 |     42.0 |
> | +CT-Module           |   36.3 |     47.3 |     76.2 |
> | +CT-Module+PWConv    |   37.2 |     48.0 |     76.7 |
> | +CT-Module+PWConv+SE |   37.2 |     48.8 |     77.4 |
> | +CT-Module+PWConv+TE |   37.3 | **50.1** | **78.8** |

---

### Official Review · AnonReviewer1 · 2020-11-01
**interesting idea, but insufficiently validated**

**Rating:** 5
**Confidence:** 4

**Review:**

This paper presents a new CNN module to learn video feature representations for action recognition, with a particular focus on increasing channel interactions for spatio-temporal modeling. To achieve that, the authors propose to divide feature channels into several sub-dimensions (called channel tensorization) and then perform group convolutions at each sub-dimension sequentially to improve channel interactions. An SE-like attention mechanism is also applied to further enhance feature representation. The proposed approach achieves competitive results on Kinetics400 and Something-Something, compared to some existing SOTA results. The paper also provides detailed ablation studies on the approach.

My understanding is that the main idea of the paper essentially performs channel shuffling followed by group convolutions at each sub-dimension. From this perspective, the idea is similar to ShuffleNet, which applies channel shuffling to enhance interactions between channel groups. While this provides interesting technical questions from the algorithmic perspective, from the point of view of the novelty, the paper does not appear as a strong technical contribution.

Another downside of the proposed approach is that it is not sufficiently validated by experiments. Firstly, the contribution of TE is not separated in Table 3 and 4, so it remains unclear whether the performance improvement is due to the enhanced channel interactions or channel attention. Secondly, the approach seems to be only effective when the number of dimensions is low, leaving it difficult to fully justify the advantages of high-dimensional channel tensorization claimed in the paper.

The FLOPs of the proposed approach in Table 4 is somewhat misleading. As indicated in the work of X3D, the performance gain from using more clips (i.e. >5x10) in evaluation is small. Nevertheless, most approaches are evaluated with 30 clips on Kinetics. I would suggest sticking to this general practice for fair comparison.

Table 3 and 4 miss some existing SOTA approaches such as X3D [1], CorrNet [3] and TAM [3], which should be listed for reference.

1) Christoph Feichtenhofer. X3d: Expanding architectures for efficient video recognition. 2020 IEEE Conference on Computer Vision and Pattern Recognition (CVPR), pp. 200–210, 2020.
2) Heng Wang, Du Tran, Lorenzo Torresani, Matt Feiszli, Video Modeling With Correlation Networks, Proceedings of the IEEE/CVF Conference on Computer Vision and Pattern Recognition (CVPR), 2020
3) Fan, Q.; Chen, C.-F. R.; Kuehne, H.; Pistoia, M.; and Cox, D. 2019. More Is Less: Learning Efficient Video Representations by Big-Little Network and Depthwise Temporal Aggregation. In Advances in Neural Information Processing Systems, 2261–2270.

---

> ### Author Response · Authors · 2020-11-23
> **Response to Reviewer 1 (Part 1)**
>
> Thanks for your constructive comments. We provide our feedbacks and modify our paper as follows:
>
> ------
>
> **Q1: The idea is similar to ShuffleNet.**
>
> **A1:** We would like to clarify the main difference.
>
> (1) **Problem.** Our CT-Net is to address video classification. In this task, we have to consider decomposition and interactions, not only along channel dimension but also spatial-temporal dimension. For this reason, we propose two principles to improve the effectiveness and efficiency of 3D CNNs, in terms of spatial, temporal and channel dimensions. This is the main contribution in this work. On the contrary, ShuffleNet works on the general image classification without insights of video modeling. Hence, it only focuses on channel shuffle and interactions in 2D CNNs.
>
> (2) **Design.** From the channel view, our CT-Net tensorizes the channel dimension as $K​$ sub-dimensions, $C=C_1\times C_2, \cdots ,\times C_K​$. This allows us to achieve sufficient channel interactions progressively, by performing convolution on each of $K​$ sub-dimensions in sequence. On the contrary, ShuffleNet only performs channel shuffle for one-time interaction.  In this case, ShuffleNet can be considered as a special case of our CT-Net with $K=2​$. From the spatial-temporal view, our CT-Net applies spatial ($1\times 3 \times3​$) and temporal ($3\times1\times1​$) convolution on each sub-dimension, which can progressively enlarge spatial-temporal receptive filed in a CT-Block for video classification. On the contrary, ShuffleNet only consists of group convolution ($1\times1​$) and depth-wise convolution ($3\times3​$), which is motivated by lightweight design in image classification.
>
> (3) **Experiment.** If we have to interpret ShuffleNet in the way of our CT operation, we perform ShuffleNet in the same backbone on Something-Something V1. The top-1 accuracy of ShuffleNet vs. CT-Net (without attention design) is **45.9%** vs. **47.3%**. It shows the superior of our CT-Net.
>
> ------
>
> **Q2: The contribution of TE is not separated in Table 3 and 4, so it remains unclear whether the performance improvement is due to the enhanced channel interactions or channel attention.**
>
> **A2:** We have made this separation in Table 2(g). Our CT-module can significantly boost its baseline, with **47.3%-16.9%=30.4%** improvement on top-1 accuracy of Something-Something V1. Our TE module can further enhance such improvement by **50.1%-48.0%=2.1%**.  It is clearly shown that our CT module makes the most important contribution to our design. Furthermore, Table 3 and Table 4 are used to show the SOTA comparison. Hence, we show our best result with the integration of CT-Module and TE.
>
> | Model                | GFLOPs |    Top-1 |    Top-5 |
> | -------------------- | :-----: | :-------: | :-------: |
> | Baseline (TSN)       |   43.0 |     16.9 |     42.0 |
> | +CT-Module           |   36.3 |     47.3 |     76.2 |
> | +CT-Module+PWConv    |   37.2 |     48.0 |     76.7 |
> | +CT-Module+PWConv+SE |   37.2 |     48.8 |     77.4 |
> | +CT-Module+PWConv+TE |   37.3 | **50.1** | **78.8** |
>
> ------
>
> **Q3:  The approach seems to be only effective when the number of dimensions is low.**
>
> **A3:** Besides accuracy, GFLOPs is also an important factor for our CT module, according to our two design principles. As shown in Table 2(b), when the number of sub-dimensions is higher, GFLOPs is smaller. It shows the effectiveness of high-dimension design in our CT-module. For example, under the comparable accuracy, the setting of $K = 4$ achieves much smaller GFLOPs than the setting of $K = 1$ (GFLOPs: **35.6** vs. **45.8**).
>
> | Number | GFLOPs | Top-1 | Top-5 |
> | ------ | :------: | :-----: | :-----: |
> | 1D     | 45.8   | 46.5  | 75.6  |
> | 2D     | 36.3   | 47.3| 76.2 |
> | 3D     | 35.7   | 47.1  | 75.8  |
> | 4D     |35.6   | 46.5  | 75.3  |
>
> ------

---

> > ### Author Response · Authors · 2020-11-23
> > **Response to Reviewer 1 (Part 2)**
> >
> > **Q4: The FLOPs of the proposed approach in Table 4 is somewhat misleading. As indicated in the work of X3D, the performance gain from using more clips (i.e. >5x10) in evaluation is small. Nevertheless, most approaches are evaluated with 30 clips on Kinetics. I would suggest sticking to this general practice for fair comparison.**
> >
> > **A4:** Thanks for your suggestion. To avoid misleading, we have updated the expressions of GFLOPs in our paper (Table 4 on page 8). There are two main reasons why we report our results using 3$\times$4 clips.
> >
> > First, for SOTA comparison, we should report the best setting of all the models, instead of the same setting. In fact, it is not appropriate to report the same setting for SOTA, due to different designs of various models. For example, SlowFast [4] uses many more frames than any other models to boost accuracy. Hence, we report our best results using 3$\times$4 clips.
> >
> > Second, the reviewer mentioned the claim that, the performance gain from using more clips (actually i.e., >1$\times$5 clips where ‘1’ means center crop) is small. This claim mainly works for X3D[1], not for all the 3D models. For example, SlowFast 8$\times$8-R101-NL [4] has 1% accuracy improvement from 1$\times$5 to 1$\times$10, as shown in Fig.3 of the X3D paper. Nevertheless, all these methods achieve the best accuracy with 3$\times$10 clips. This is why they report 3$\times$10 clips for SOTA comparison. Alternatively, it is not quite applicable to our CT-Net. In fact, the 3$\times$10 setting of our CT-Net is slightly worse than the 3$\times$4 setting (top-1 accuracy of 3$\times$10 vs. 3$\times$4: 78.7% vs. 78.8%). Hence, we report our CT-Net with its best accuracy setting. This is the same best-accuracy strategy as other methods.
> >
> > ------
> >
> > **Q5:  Table 3 and 4 miss some existing SOTA approaches such as X3D [1], CorrNet [2] and TAM [3], which should be listed for reference.**
> >
> > **A5:** Thanks for these references. We have added these discussions in the updated paper (Table 3 on page 7 and Table 4 on page 8). First, our CT-Net outperforms CorrNet [2] and TAM [3]. For example, our CT-Net-50 is better than CorrNet-50 [2] on Something-Something V1 (top-1 accuracy: 51.7% vs 49.3%). It also outperforms TAM [3] on Kinetics-400 (top-1 accuracy: 77.3% vs. 72.0%). Second, we can achieve comparable top-1 accuracy with X3D [1] (our CT-Net-R101 vs. X3D-XL: 78.8% vs. 79.1%), under the similar GFLOPs (our CT-Net-R101 vs. X3D-XL: 1746 vs. 1452). However, X3D requires extensive model searching with an expensive GPU setting, while our CT-Net can be trained traditionally with feasible computation.
> >
> > ------
> >
> > [1] Christoph Feichtenhofer. X3d: Expanding architectures for efficient video recognition. 2020 IEEE Conference on Computer Vision and Pattern Recognition (CVPR), pp. 200–210, 2020.
> >
> > [2] Heng Wang, Du Tran, Lorenzo Torresani, Matt Feiszli, Video Modeling With Correlation Networks, Proceedings of the IEEE/CVF Conference on Computer Vision and Pattern Recognition (CVPR), 2020.
> >
> > [3] Fan, Q.; Chen, C.-F. R.; Kuehne, H.; Pistoia, M.; and Cox, D. 2019. More Is Less: Learning Efficient Video Representations by Big-Little Network and Depthwise Temporal Aggregation. In Advances in Neural Information Processing Systems, 2261–2270.
> >
> > [4] Christoph Feichtenhofer, Haoqi Fan, Jitendra Malik, and Kaiming He. Slowfast networks for video recognition. 2019 IEEE/CVF International Conference on Computer Vision (ICCV), pp. 6201– 6210, 2019.

---

### Decision · Program_Chairs · 2021-01-07
**Final Decision**

**Decision:**

Accept (Poster)

**Comment:**

The paper focuses on the task of learning efficient representation models for video classification. To avoid the excessive computational cost of performing 3D convolutions on video, the authors propose to break the channel dimension of video representations into sub-dimensions that are treated separately. This cuts down on computation and improves classification performance over many methods in the literature. Extensive experiments were run on well-known benchmarks to justify the claims of the model. Such backbone architectures can be very useful in the realm of video understanding. The authors should be commended for the amount of work they did in the rebuttal period to address the comments and inquiries brought up by the reviewers. Extra experiments were done and more in-depth analysis was made possible.